# Stress and benefits of video calling for people with autism spectrum disorders

**Kengo Yuruki**[◐], **Masahiko Inoue**[iD]*[◐]

Department of Clinical Psychology, Graduate School of Medical Sciences, Tottori University, Yonago, Japan

◐ These authors contributed equally to this work.
* masahiko-inoue@tottori-u.ac.jp

## Abstract

This study compared stress and the benefits of video calling between individuals with autism spectrum disorder (ASD) proneness and diagnosis and those with typical developmental (TD). Study participants were recruited via the web, and 151 of the 252 participants who responded to a web-based questionnaire were included in the analysis (ASD group:76; TD group:75). The results of the chi-square test suggest that the ASD group may prefer video calling more than the TD group. The results of the analysis using a qualitative methodology (KJ method) suggested that the ASD group was more likely than the TD group to perceive stress due to light stimuli emitted from the screen and the inability to concentrate on a conversation due to visual stimuli. The ASD group perceived the ability to cope with stressful stimuli by operating the device as an benefits of video calling. These findings suggest the importance of creating a communication environment that reduces stress and maximizes the benefits of video calling for people with ASD. Specific support measures include establishing rules in advance that allow the individual to turn the video off or switch to texting.

**Data Availability Statement:** All relevant data are within the paper and its Supporting Information files.

**Funding:** The authors received no specific funding for this work.

## Introduction

Autism spectrum disorder (ASD) is a neurodevelopmental disorder whose main diagnostic features are impairments in reciprocal social communication and interpersonal interaction; limited and repetitive behaviors, interests, or modes of activity; and hypersensitivity to sensory stimuli [1]. The core diagnostic features of ASD and other characteristics can significantly impact face-to-face communication among individuals with ASD. For example, hypersensitivity to sensory stimuli has been reported to cause overload from loud sounds, strong smells, bright lights, and so on, threatening social skills and making face-to-face interactions difficult [2, 3]. Individuals with ASD may take longer to respond because they need to go through many information-processing processes before deciding what to say and do, have difficulty understanding complex communication due to attention-switching issues, and may have difficulty making eye contact with the other person and interpreting their facial expressions [2–4].

The difficulties individuals with ASD may have in face-to-face communication can be alleviated by text-based communication over the Internet. Text-based communication, such as e-mail, is highly valued by individuals with ASD because it gives them additional time to think

**Competing interests:** The authors have declared that no competing interests exist.

of their responses, there is no pressure for eye contact [2, 5], they can communicate at their own pace, and there is no nonverbal communication [6].

In recent years, there has been a growing need for Internet communication using video calling mediums such as Skype. Video calling is an Internet-based technology that provides a synchronous experience of seeing another person and hearing their voice [7]. Video calling is different from traditional forms such as face-to-face communication or email; it is characterized by multitasking (which involves operating devices in addition to speaking) [8], gaps in eye contact between speakers due to camera position [9], delays in communication [10], and fatigue [11]. These characteristics differ significantly from text-based internet communication, which has already been studied in individuals with ASD.

Only a few studies have been conducted on video calling communication among individuals with ASD. Those studies suggest that individuals with ASD experience difficulties in video calling [12, 13]. Morris et al. [12] found that, concerning forms of communication in the workplace, neurodiverse groups, such as those with ASD and ADHD, were more likely to rate video calling communication as uncomfortable than those with typical developmental (TD). However, as Morris et al. [12] did not examine this in detail, it is unclear why the group with ASD was more likely to rate video calling as unpleasant. Zolyomi et al. [13] found that individuals with ASD were stressed about three aspects of video calling communication: sensory sensitivity, cognitive load, and anxiety; however, only individuals with ASD were included in their study, and therefore, it is not clear whether the stress revealed is specific to individuals with ASD. Therefore, previous studies have not examined stressors through a direct comparison between individuals with ASD and TD, and the stressors that individuals with ASD can characteristically experience in video calling are not yet clear.

Although this study focuses on the negative aspects of video calling in individuals with ASD, video calling may have positive aspects for individuals with ASD. Zolyomi et al. [13] showed that, while people with ASD experience stress during video calling, there is potential for flexible coping strategies by way of application and device features such as adjusting screen color temperature and volume and being able to turn the video off. Therefore, video calling may be more beneficial than other forms of communication for individuals with ASD for coping with various stressors. However, the positive aspects of video calling, as perceived by individuals with ASD, have not been explored.

In this study, we examined the perceived stress of video communication among individuals with ASD proneness or diagnosis, and the benefits of video communication by considering the perceived benefits and preferences for video communication. The three research questions of this study are as follows: RQ1: Are there differences between individuals with ASD and TD in their preferences for video calling and face-to-face communication? RQ2: What differences exist in the perceived stress of video calling between individuals with ASD and TD? RQ3: What differences exist in the perceived benefits of video calling between individuals with ASD and TD?

Qualitative analysis methods were used to examine the perceptions of stress and benefits mentioned in RQs 2 and 3. At present, it is unclear what aspects of stress and benefits are perceived by people with ASD proneness and diagnosis during video calling. Therefore, a qualitative study based on the subjective perceptions of the participants was conducted to clarify these aspects in an exploratory manner. In addition, to better clarify the stressors and benefits of video calling for individuals with ASD proneness and diagnosis, this study was conducted in comparison with individuals with TD. In qualitative research, attempts have been made to compare the results of analyses using comparison groups. Such attempts can show similarities and differences by comparing different groups and aspects of the phenomenon, highlighting areas in need of further support [14]. Therefore, we hypothesized that, by comparing the

stressors and benefits of video calling in individuals with ASD proneness and diagnosis to those with TD, similarities and differences can be observed between the two groups. This attempt may also allow for a more detailed examination of video calling and support for it among those with ASD proneness and diagnosis.

The purpose of this study was to qualitatively analyze open-ended responses obtained through an Internet-based questionnaire survey regarding communication form preferences, stress, and perceived benefits of video calling among individuals with ASD proneness or diagnosis, and those with TD. It also examines the kind of support needed to reduce stress and maximize the benefits of video calling in individuals with ASD.

## Materials and methods

### Participants

In this study, participants were recruited by sending an e-mail to ASD patient organizations and through Twitter, a social networking service. For recruitment by e-mail to ASD organizations, the URL and QR codes of a Google form containing the outline of this study and the questionnaire was sent through e-mail. For recruitment via Twitter, the second author's account providing information was used to post an overview of the study and the URL of the dedicated web page.

Those redirected to the web page from the URL were given a more detailed overview of the study. If they agreed to cooperate with the study after reading the study outline, they were instructed to go to Google Forms. Two Google Forms were created: one for individuals with ASD proneness or diagnosis and another for individuals with TD. Participants who accessed the web page through the URL selected one or the other by themselves. For both recruitment methods, before answering the questions, we explained that participation in the study was voluntary, that personal information would be kept confidential, that the results of the study might be used for academic purposes, and that no personally identifiable information would be made public in such cases. Only those who consented to participate in the study were allowed to answer the questionnaire.

A total of 252 participant responses were obtained (130 from the form for those with ASD proneness and diagnosis and 122 from the form for individuals with TD). Of these, we screened those eligible for analysis based on their scores on the Autism-Spectrum Quotient Japanese Version 16 (AQ-J-16; [15]), presence of ASD proneness or diagnosis, and mental disorders. Fig 1 shows the screening procedure.

### Autism-Spectrum Quotient Japanese Version 16 (AQ-J-16)

The Autism-Spectrum Quotient (AQ) is an instrument for rapidly quantifying where any given individual is situated on the continuum from autism to normality [16]. In this study, we used the AQ-J-16, a short-form of the Japanese version of the AQ, the Autism Spectrum Quotient-J (AQ-J) [17]. The item numbers extracted from the AQ for the short version were 7, 15, 18, 20, 24, 26, 27, 31, 32, 34, 35, 39, 41, 42, 45, and 46. While the reliability and validity of the AQ-J have been confirmed [17], those of the AQ-J-16 have not been [15]. However, the sensitivity (0.80), specificity (0.97), and positive (0.80) and negative predictive value (0.97) of the AQ-J-16 have been reported to be superior to and more useful than the AQ-J in screening for ASD [15]. When answering the AQ-J-16, respondents were asked to consider their current situation rather than recall the past. The AQ-J-16 was rated on a four-point scale ("1: *definitely different*," "4: *definitely similar*"), and the cut-off point was 12.

The participants had to meet the following conditions to be included in the analysis: 1) those with an AQ-J-16 score above the cutoff and a diagnosis of ASD or ASD proneness and 2)

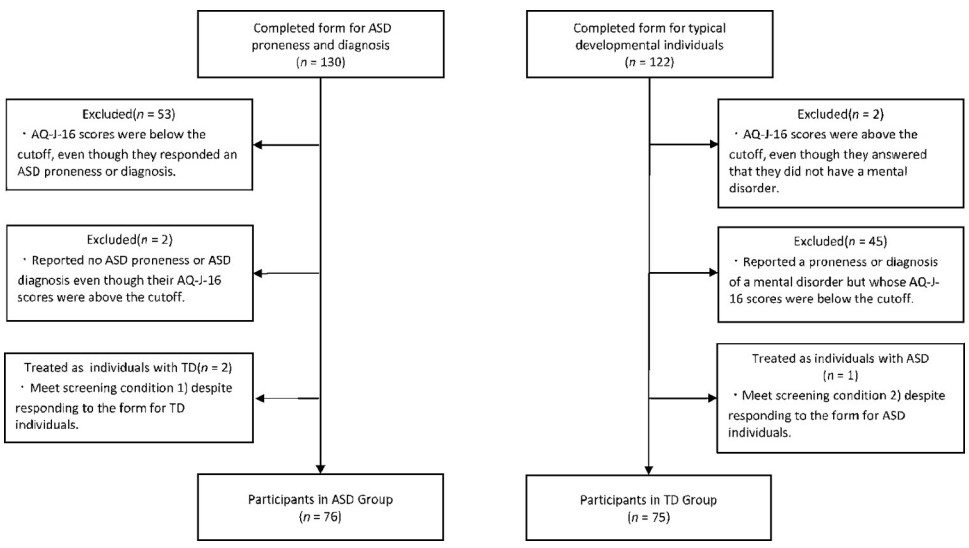

**Fig 1. Screening flow.**

those with an AQ-J-16 score below the cutoff and no diagnosis or proneness of any mental disorder, including ASD. Based on these conditions, 53 individuals whose AQ-J-16 scores were below the cutoff on the form for ASD proneness and diagnosis were excluded from the analysis even though they responded that they had ASD proneness or diagnosis.

Two individuals who reported no ASD proneness or diagnosis even though their AQ-J-16 scores were above the cutoff were also excluded from the analysis. Two individuals with TD whose AQ-J-16 scores were above the cutoff were excluded from the analysis even though they answered that they did not have a proneness or diagnosis of a mental disorder, including ASD. In addition, 45 individuals who reported proneness or diagnosis of a mental disorder but whose AQ-J-16 scores were below the cut-off for individuals with TD were excluded from the analysis.

Participants who met Condition 1 despite responding to the form for individuals with TD were treated as analysis participants with ASD proneness or diagnosis. Participants who met Condition 2 despite answering the form for ASD proneness and diagnosis, were treated as analysis participants for TD. As a result, one participant from the form for TD was considered an individual with ASD proneness and diagnosis, and two participants from the form for ASD proneness or diagnosis were considered individuals with TD.

After screening, 76 individuals with ASD proneness and diagnosis (14 male, 59 female, and three others) and 75 individuals with TD (15 male, 58 female, and two others) were included in the analysis. The participants were divided into two groups: ASD and TD. Table 1 presents the characteristics of the participants in each group.

## Questionnaire

A questionnaire was created using Google Forms. It consisted of demographic items such as age, sex, and mental disorder status of the participants; the AQ-J-16; items about preferences regarding the form of communication; and open-ended items about the stress and benefits of video calling. The same items were used for the ASD and TD groups. As the open-ended items were considered highly abstract, examples of responses were provided to reduce the difficulty of answering the questions. Before answering the open-ended item, the participants were asked, "Do you talk more often on one-on-one or multi-person video calling?" Based on the

**Table 1. Characteristics of the participants.**

|  | ASD Group(*n* = 76) | TD Group(*n* = 75) |
| --- | --- | --- |
| **Sex(%)** |  |  |
| **Male** | 14(18.42) | 15(20.00) |
| **Female** | 59(77.63) | 58(77.33) |
| **Other** | 3(3.95) | 2(2.67) |
| **Average age(*SD*)** | 35.62(9.30) | 36.57(9.65) |
| **AQ-J-16 score** | 13.53(1.06) | 8.03(1.72) |
| **Proneness/Diagnosis(%)** |  |  |
| **ASD only** | 8(10.53) |  |
| **ADHD comorbidity** | 54(71.05) |  |
| **LD comorbidity** | 8(10.53) |  |
| **SAD comorbidity** | 11(14.47) |  |
| **Other mental disorder comorbidity** | 29(38.16) |  |
| **Other disorder comorbidity** | 2(2.63) |  |

AQ-J-16, autism-spectrum quotient Japanese version 16; ASD, autism spectrum disorder; ADHD, attention deficit hyperactivity disorder; LD, learning disorder; SAD, social anxiety disorder.

answers to this question, the participants were asked to think of situations in which they experienced more video calling.

## Data analysis

For the item assessing the preference of conversation format, we calculated the selection rates for video calling and face-to-face conversations in the ASD and TD groups, and conducted a chi-square test. In conducting the chi-square test, based on previous studies, we considered that people with ASD may be more influenced by sensory stimuli during video calling. Therefore, we hypothesized that the ASD group would be more likely than the TD group to choose face-to-face conversations and less likely to choose video calling. Chi-square tests were conducted using IBM SPSS Statistics version 28 for Windows.

In this study, the open-ended responses obtained from the ASD and TD groups were analyzed using the KJ method developed by cultural anthropologist Jiro Kawakita [18], which is a major analytical method for qualitative research in Japan [19]. It involves compiling collected qualitative data and creating a new meaning system, emphasizing the idea of new meanings found by combining data [18, 19].

The procedure for the analysis based on the KJ method in this study is as follows: first, we extracted meaningful cohesion (segments) from the open-ended responses. Segments were extracted such that each segment contained only one meaning. If more than one meaning was included, each segment was divided into another segment. Second, all extracted segments were examined multiple times, and segments with similar meanings were grouped into one category. Third, a summary that successfully expressed the common elements of all the segments belonging to each category was prepared. The summary was then used as the name of each category. Based on the category names, we then created higher-order categories by grouping categories highly similar to each other. These steps were repeated until no more categories were deemed similar.

## Ethical consideration

This study was approved by the Ethical Review Committee of Tottori University School of Medicine (approval number:21A051).

# Results

## Usage of video calling

Table 2 shows the use of video calling in the ASD and TD groups.

## Preferred communication mode

To determine the preferred communication mode (video or face-to-face conversation), we performed a chi-square independence test and residual analysis by calculating the selection rates of the two communication modes in the ASD and TD groups. Table 3 shows the results of the chi-square test for independence and the residual analysis. The chi-square test of independence showed that the chi-square value was significant ($\chi^2(2) = 10.79$, $p = .001$). Therefore, a residual analysis was subsequently performed to examine the differences in selection rates between the groups in detail. The results of the residual analysis showed that the adjusted residual standardized values for each selection rate in the two groups exceeded ±1.96, indicating a significant difference in frequency. These results indicate that the selection rates of video calling in the ASD group and face-to-face conversations in the TD group were significantly higher than other selection rates, and the hypothesis that the ASD group would select face-to-face conversations more often was not supported.

## Analysis of open-ended question

The KJ-method analysis revealed 280 segments in the ASD group (149 for stress and 131 for benefits of video calling) and 290 segments in the TD group (152 for stress and 138 for benefits of video calling). The final number of categories was 125 for the ASD group (69 for stress and 56 for benefits of video calling) and 120 for the TD group (63 for stress and 57 for benefits of video calling). The categories ranged from the lowest-order first category, formed by combining segments, to the highest-order fourth category, formed by combining categories.

**Table 2. Use of video calling in the ASD and TD groups.**

|  | ASD Group($n = 76$) | TD Group($n = 75$) |
|---|---|---|
| **Applications in use(%)** |  |  |
| **Zoom** | 64(84.21) | 69(92.00) |
| **Skype** | 19(25.00) | 6(8.00) |
| **Google Meet** | 12(15.79) | 23(32.67) |
| **Other** | 27(35.53) | 25(33.33) |
| **Devices in use(%)** |  |  |
| **Desktop** | 7(9.21) | 6(8.22) |
| **Laptop** | 43(56.58) | 59(80.82) |
| **Tablet** | 9(11.84) | 2(2.74) |
| **Smartphone** | 17(22.37) | 6(32.88) |
| **Frequency of use(%)** |  |  |
| **Once or twice a month** | 42(55.26) | 30(41.1) |
| **Three to Four times a month** | 10(13.16) | 8(10.96) |
| **Five to Six times a month** | 12(15.79) | 6(32.88) |
| **More** | 12(15.79) | 29(39.73) |
| **Conversational situations(%)** |  |  |
| **One-on-one** | 20(26.32) | 17(23.29) |
| **Multi-people** | 56(73.68) | 56(76.71) |

**Table 3. Results of the chi-square test of independence and residual analysis.**

| | Video Calling(*n* = 47) | Face-to-face(*n* = 104) |
|---|---|---|
| | *n*(%) | *n*(%) |
| | *sr* | *sr* |
| **ASD Group** | 33(43.4) | 43(56.6) |
| | 3.3 | 3.3 |
| **TD Group** | 14(23.3) | 61(81.3) |
| | 3.3 | 3.3 |

## Stress in video calling

Fig 2 shows the results of the KJ method analysis of stress in video calling. Fig 2 show some of the results of the analysis from the fourth to the second categories. The categories of stress in video calling were considered to be similar between the two groups. For example, categories

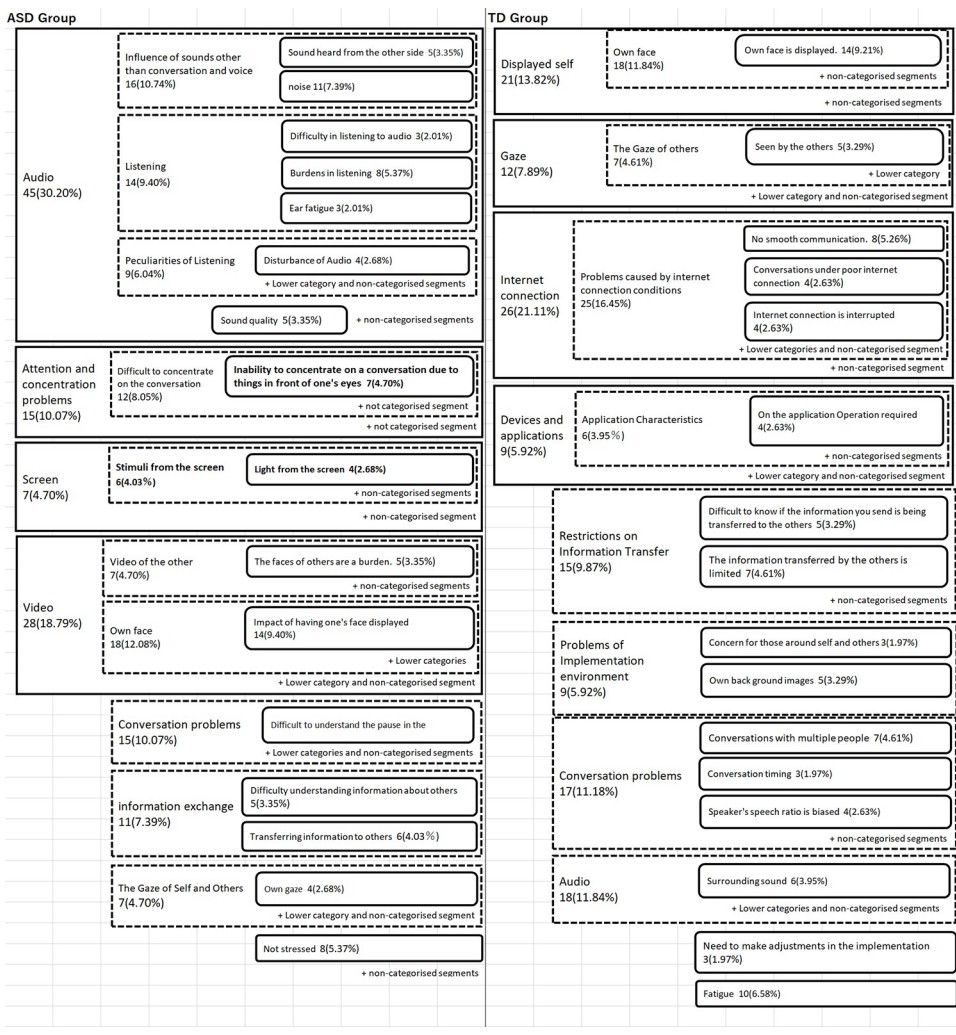

**Fig 2. Results of KJ method analysis on the stress of video calling.** Bolded category names indicate categories considered to have been shown to be characteristic of the ASD Group based on a comparison of the two groups. The numbers indicate the number of segments that mentioned the content belonging to the category, and the numbers in parentheses indicate the percentage of the number of all segments in each group.

indicating stress during information transmission (e.g., "Information exchange") and stress due to the display of one's face and others' face (e.g., "Video") were commonly indicated. The category of stress related to sound ("Audio") was also common in both groups, but the TD group had only 11.84% of the total intercepts corresponding to this category, whereas the ASD group had 30.20% of the total intercepts, indicating a gap between the two groups. The subcategory "Inability to concentrate on a conversation due to things in front of one's eyes," which constituted the "Attention and concentration problems" category, was characteristic of the ASD group. The subcategories of "Stimuli from the screen" and "Light from the screen," which constitute the category of "Screen," were also characteristic of the ASD group.

## The benefit of video calling

Fig 3 shows the results of the analysis of the benefits of video call Fig 3 show some of the results of the analysis from the fourth to the second categories. Many categories of benefits of video calling were common to both groups. For example, in the category of "Benefits of not being face-to-face," more than half of all segments in both groups (63.36% in the ASD group and 57.25% in the TD group) showed the corresponding content. In contrast, the ASD group showed characteristic content in the lower categories. For example, the second category, "Less influenced by the other person's visual information," which was included in the category of "Benefits of not being face-to-face," was found only in the ASD group. In addition, the category of "Can deal with problems in one's own time using application functions" included in

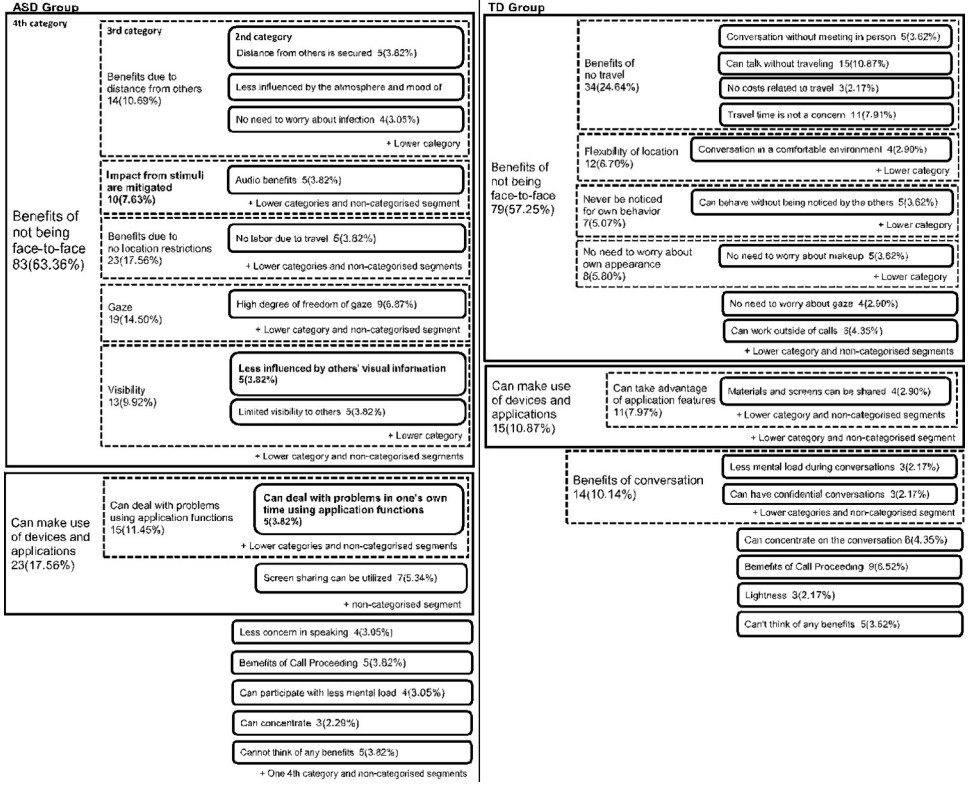

**Fig 3. Results of KJ method analysis on the benefits of video calling.** Bolded category names indicate categories shown to be characteristic of the ASD Group based on a comparison of the two groups. The numbers indicate the number of segments that mentioned the content belonging to the category, and the numbers in parentheses indicate the percentage of the number of all segments in each group.

the category of "Can make use of devices and applications" was also found to be characteristic only in the ASD group.

## Discussion

This study investigated the perceived preference, stress, and benefits of video calling in communication between individuals with ASD proneness or diagnosis, and those with TD. We then compared individuals with ASD proneness or diagnosis and those with TD using the KJ method and other analytic methods.

### Preferred communication mode

The results of the chi-square test showed a significant difference between the ASD and TD groups in their preferences for each form of communication. In other words, the ASD group was more likely to prefer video calling than the TD group. This result is in contrast with Morris et al. [12], who found that the ASD group was more likely to rate video calling as unpleasant than the TD group. The preference for video calls may be related to the perceived benefits.

Currently, owing to the prevalence of coronavirus disease 2019 (COVID-19), the use of remote tools, including video calling, is under recommendation, especially in Japan [20]. Therefore, compared to the participants in Morris et al. [12], those in our study may have a higher affinity for video calling and are more likely to recognize its benefits and preferences. However, Morris et al. [12] targeted communication in a work environment and their perceptions of communication may be unique. Therefore, a simple comparison with the results of this study cannot be made.

### Stress in video calling

Stress in video calling in the ASD group was characterized by the categories of "Stimuli from the screen," "Light from the screen," and "Inability to concentrate on the conversation due to what is in front of the eyes." Stress due to being influenced by visual stimuli was more characteristic in the ASD than the TD group. Stress due to "Attention and concentration problems" were found in both groups, but the ASD group had a higher level of this category. Therefore, it can be inferred that this topic was perceived as more stressful by the ASD group. In the ASD group, the subcategory of "Attention and concentration problems" included "Inability to concentrate on a conversation due to things in front of one's eyes." Therefore, visual stimuli may have affected the stress of not being able to concentrate in the ASD group.

Stress related to "Audio" was also a common category in both groups. However, it accounted for 11.84% of the total segments in the TD group, whereas it accounted for 30.20% of the total segments in the ASD group. This may indicate that the ASD group was more likely to perceive stress related to the speech aspects. According to Zolyomi et al. [13], the stress of sensory oversensitivity related to auditory and other stimuli is associated with video calling in individuals with ASD. In the present study, it was found that stress from light stimuli emitted from the screen, inability to concentrate on the conversation due to visual information displayed on the video calling, and stress related to audio on the video calling may be more easily perceived by the ASD than the TD group.

These data indicate that individuals with ASD diagnoses and tendencies can have unique stressors in video calling, suggesting the need for support for stressors in video calling in individuals with ASD.

## Benefits of video calling

Regarding the perceived benefits of video calling, analyzed qualitatively using the KJ method, "Benefits of not being face-to-face" and "Can make use of devices and applications" were the main common factors between the ASD and TD groups.

However, an examination of the lower categories revealed that some of them were only found in the ASD group. For example, in the ASD group, among the subcategories included in the category "Benefits of not being face-to-face," there was a category "Impact from stimuli are mitigated." However, this category was not observed in the TD group. In addition, the ASD group's category of "Can make use of devices and applications" included the category of "Can deal with problems in one's own time using application functions." This category is unique to the ASD group. Although the use of applications and device operation were commonly observed in TDs, the ASD group may have more actively used applications and device operation to cope with problems, including stress, and recognized the benefits in this respect.

Individuals with ASD may have difficulty interacting with each other in face-to-face communication environments due to the influence of light stimuli (e.g., [2]). We found that similar stimuli exist in the video conference environment, and that they can affect individuals with ASD as stressors. However, in a video calling environment, these stimuli can be cut by manipulating the equipment, which may render video calling a comfortable communication environment for people with ASD. Support should be provided to maximize the benefits perceived by the parties involved in the video calling environment.

In addition, Morris et al. [12] showed that groups with ASD tended to rate video calling communication as unpleasant compared to the TD group, but the findings of this study were different. As mentioned earlier, Morris et al.'s [12] study is limited to communication in a work environment, and thus cannot be simply compared to the results found in the present study. However, those with ASD who perceive the ability to proactively deal with burdens as a benefit of video calling, such as the ASD group in this study, may tend to prefer video calling to face-to-face communication.

## Assistance needed in video calling with persons with Autism Spectrum Disorders

Support during implementation in video calling environments has been considered in the medical context (e.g., [21–23]). Bischoff et al. (2004) found that, if the impact of technology on communication is not addressed, interruptions and the resulting frustration can exacerbate the therapeutic relationship. Bischoff et al. [21] suggested addressing the communication process in remote treatment settings (e.g., clearer body language, changing pace and timing, and conducting shortened versions of sessions or telephone sessions in case of equipment failure or poor connections).

However, no studies have examined how to support individuals with ASD in video calling. In this study, the benefit of video calling was recognized in the ASD group as the ability to respond to load by operating the equipment. Therefore, it is important to support the creation of a comfortable communication environment for people with ASD. Specifically, it is to guarantee the opportunity to take coping actions through the functionality of the equipment or application. For example, it is possible to establish rules in advance that allow the user to, depending on the load generated, turn the screen off or switch to chatting as appropriate.

In recent years, there have been attempts to apply video calling to remote assistance such as medical services [24], known as telehealth [25]. Cognitive-behavioral therapy and coaching via video calling have been reported for individuals with ASD [8, 26]. With the current situation of COVID-19, the importance of providing services remotely in terms of social distancing has

been recognized [27], and the use of video calling is likely to increase in the future. In this context, it is important to create a communication environment that reduces stress and maximizes the benefits that patients with ASD may experience during video calling.

## Limitations

The present study has some limitations. First, the ASD group in the present study included both those with a diagnosis and those with only proneness but no diagnosis; therefore, caution is needed in interpreting the results. Second, most participants in the ASD group in this study had comorbid mental disorders, such as ADHD, LD, and social anxiety. As video calling may cause experiences related to the characteristics of disorders such as social anxiety [28], caution should be exercised when interpreting the results of the present study. In the future, knowledge on the experiences of video calling related to ASD and other disabilities should be accumulated through more rigorous screening and other means. Third, the age of participants and frequency of video calling use may confounded the results of the study. Younger age is a factor that increases the frequency of video calling use [29], and furthermore, frequency of use affects fatigue on video calling [11]. Therefore, the confounding of participants' age may create a bias in the frequency of video calling use and change the perception of stress in video calling. In the future, analyzing the differences in experience according to age and frequency of video calling use. The fourth point is the possible influence of the instructions provided to answer the questions. In this study, the respondents were instructed to answer questions regarding situations with which they had more frequent experiences, whether it was a conversation with several people or one-on-one. Therefore, the results of this study may include a combination of responses based on one-on-one and multiple situations. In the future, a detailed study assessing respondents' stress in each situation should be conducted. Finally, as this study was based primarily on qualitative analysis, it is difficult to generalize the results of this study broadly. In the future, quantitative studies on various samples based on the findings of this study should be conducted.

## Conclusion

The present study qualitatively compared preferences of communication mode and stress and benefits in video calling between individuals with ASD proneness or diagnosis and individuals with TD. The results indicated that the ASD group was more likely than the TD group to prefer to talk over video calling. The ASD group also showed stress due to light stimuli emitted from the screen and stress due to visual stimuli specific to video calling, such as the display of one's face, which made it difficult to concentrate on the conversation. In contrast, the ASD group recognized the ability to cope with stressors by operating the device as a benefit. These results suggest the importance of creating a communication environment that reduces stress and maximizes benefits for individuals with ASD.

## Supporting information

**S1 Dataset. Minimal underlying dataset.**
(XLSX)

## Acknowledgments

We thank all participants for their cooperation in this study.

## Author Contributions

**Conceptualization:** Masahiko Inoue.

**Data curation:** Kengo Yuruki.

**Formal analysis:** Kengo Yuruki.

**Investigation:** Kengo Yuruki.

**Methodology:** Masahiko Inoue.

**Project administration:** Masahiko Inoue.

**Resources:** Kengo Yuruki.

**Supervision:** Masahiko Inoue.

**Writing – original draft:** Kengo Yuruki.

**Writing – review & editing:** Masahiko Inoue.

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
