## [Decision Letter · Decision Letter 0]

16 Jan 2023

PONE-D-22-26680Stress and benefits of video calling for people with autism spectrum disordersPLOS ONE

Dear Dr. INOUE,

Thank you for submitting your manuscript to PLOS ONE. After careful consideration, we feel that it has merit but does not fully meet PLOS ONE’s publication criteria as it currently stands. Therefore, we invite you to submit a revised version of the manuscript that addresses the points raised during the review process.

ACADEMIC EDITOR:

Please respond to the small methodological questions raised by the reviewers.Thank you.

We look forward to receiving your revised manuscript.

Kind regards,

Santiago Gascón, PhD

Academic Editor

PLOS ONE

Journal Requirements:

3. Please upload a copy of Supporting Information Figures and tables which you refer to in your text on page 38.

Reviewers' comments: 

Reviewer's Responses to Questions

**Comments to the Author**

1. Is the manuscript technically sound, and do the data support the conclusions?

Reviewer #1: Yes

2. Has the statistical analysis been performed appropriately and rigorously? 

Reviewer #1: Yes

3. Have the authors made all data underlying the findings in their manuscript fully available?

Reviewer #1: No

4. Is the manuscript presented in an intelligible fashion and written in standard English?

Reviewer #1: Yes

5. Review Comments to the Author

Reviewer #1: In methodology there was not mention about age limits. Age is an important indicator for coping any new application. In different age group person's ability to minimise stress at different level, like at early age group ASD people could cope early. Time of screen time also an important issue which author mentioned in limitation.

6. PLOS authors have the option to publish the peer review history of their article (what does this mean?). If published, this will include your full peer review and any attached files.

Reviewer #1: No

---

## [Author Response · Author response to Decision Letter 0]

9 Mar 2023

We thank the reviewers for their valuable comments that have greatly improved our paper.

Comment:

３．Have the authors made all data underlying the findings in their manuscript fully available?

Response: We have added a Minimal Underlying dataset file to the Data Availability statement.

Comment: In methodology there was not mention about age limits. Age is an important indicator for coping any new application. In different age group person's ability to minimise stress at different level, like at early age group ASD people could cope early. Time of screen time also an important issue which author mentioned in limitation.

Response: Thank you for pointing this out. We have changed the following text in “Limitations” from (p. 30, lines 407-410): 

Third, the frequency of video calling may have confounded the results. In particular, the results of video calling fatigue may differ depending on the call frequency [11]. In the future, analyzing the differences in experience according to frequency of use would be desirable.

to 

Third, the age of participants and frequency of video calling use may confounded the results of the study. Younger age is a factor that increases the frequency of video calling use [29], and furthermore, frequency of use affects fatigue on video calling [11]. Therefore, the confounding of participants’ age may create a bias in the frequency of video calling use and change the perception of stress in video calling. In the future, analyzing the differences in experience according to age and frequency of video calling use.

Ｗe have also added the following reference.

29. Brown G, Greenfield PM. Staying connected during stay-at-home: Communication with family and friends and its association with well-being. Hum Behav Emerging Technol. 2021;3(1): 147-156. doi: 10.1002/hbe2.246.

We wish to thank the Reviewer again for his or her valuable comments.

---

## [Editor Report · Decision Letter 1]

12 Mar 2023

Stress and benefits of video calling for people with autism spectrum disorders

PONE-D-22-26680R1

Dear Dr. Inoue,

We’re pleased to inform you that your manuscript has been judged scientifically suitable for publication and will be formally accepted for publication once it meets all outstanding technical requirements.

Kind regards,

Santiago Gascón, PhD

Academic Editor

PLOS ONE
---

## [Editor Report · Acceptance letter]

11 Apr 2023

PONE-D-22-26680R1 

Stress and benefits of video calling for people with autism spectrum disorders 

Dear Dr. Inoue:

I'm pleased to inform you that your manuscript has been deemed suitable for publication in PLOS ONE. Congratulations! Your manuscript is now with our production department. 

Kind regards, 

on behalf of

Dr. Santiago Gascón 

Academic Editor

PLOS ONE